# GDE7 produces cyclic phosphatidic acid in the ER lumen functioning as a lysophospholipid mediator

Keisuke Kitakaze [1,6✉], Hanif Ali[2], Raiki Kimoto[1,3], Yasuhiro Takenouchi[1], Hironobu Ishimaru[1], Atsushi Yamashita[4], Natsuo Ueda[5], Tamotsu Tanaka[2], Yasuo Okamoto[1] & Kazuhito Tsuboi [1,6✉]

Cyclic phosphatidic acid (cPA) is a lipid mediator, which regulates adipogenic differentiation and glucose homeostasis by suppressing nuclear peroxisome proliferator-activated receptor γ (PPARγ). Glycerophosphodiesterase 7 (GDE7) is a $Ca^{2+}$-dependent lysophospholipase D that localizes in the endoplasmic reticulum. Although mouse GDE7 catalyzes cPA production in a cell-free system, it is unknown whether GDE7 generates cPA in living cells. Here, we demonstrate that human GDE7 possesses cPA-producing activity in living cells as well as in a cell-free system. Furthermore, the active site of human GDE7 is directed towards the luminal side of the endoplasmic reticulum. Mutagenesis revealed that amino acid residues F227 and Y238 are important for catalytic activity. GDE7 suppresses the PPARγ pathway in human mammary MCF-7 and mouse preadipocyte 3T3-L1 cells, suggesting that cPA functions as an intracellular lipid mediator. These findings lead to a better understanding of the biological role of GDE7 and its product, cPA.

[1] Department of Pharmacology, Kawasaki Medical School, Kurashiki, Okayama, Japan. [2] Graduate School of Technology, Industrial and Social Sciences, Tokushima University, Tokushima, Japan. [3] Nara Medical University, Kashihara, Nara, Japan. [4] Laboratory of Biological Chemistry, Faculty of Pharma-Science, Teikyo University, Tokyo, Japan. [5] Department of Biochemistry, Kagawa University School of Medicine, Miki, Kagawa, Japan. [6] These authors jointly supervised this work: Keisuke Kitakaze, Kazuhito Tsuboi. ✉email: kitakaze@med.kawasaki-m.ac.jp; ktsuboi@med.kawasaki-m.ac.jp

Lysophosphatidic acid (LPA) and cyclic phosphatidic acid (cPA) are lipid mediators in eukaryotic tissues and plasma[1,2]. LPA regulates development, physiological functions, and pathological processes[3] through G protein-coupled receptors[4]. LPA is mainly extracellularly produced from lysophosphatidylcholine (LPC) and other lysophospholipids by autotaxin (ATX) and is degraded by a class of lipid phosphatases[4]. Meanwhile, cPA and its synthetic analogs, carbacPAs, inhibit cancer invasion and metastasis through ATX inhibition[5], and promote hyaluronic acid biosynthesis via LPA receptor activation[6]. LPA and cPA also modulate adipogenic differentiation and glucose homeostasis through nuclear peroxisome proliferator-activated receptor γ (PPARγ, encoded by *PPARG*) as an agonist and an antagonist, respectively[7,8]. Although the biosynthetic mechanisms of LPA[3,7] and another endogenous agonist, 15-deoxy-$\Delta^{12,14}$-prostaglandin $J_2$[9,10], are well established, those of cPA are not fully elucidated in vivo.

cPA is synthesized in mammalian cells by ATX[11] and phospholipase D2 (PLD2)[8]. ATX is a lyso-PLD-type exoenzyme in the plasma, which hydrolyzes various lysophospholipids, including LPC, lysophosphatidylethanolamine, and lysophosphatidylserine to produce LPA[11]. ATX also possesses transphosphatidylation activity toward LPC to form cPA, although this activity is much lower than its hydrolysis activity to generate LPA[12]. Meanwhile, PLD2 mainly hydrolyzes phosphatidylcholine to phosphatidic acid in the cell membrane[13] and also produces LPA and cPA from LPC[8].

Recent evidence indicates that some members of the glycerophosphodiesterase (GDE) family have phospholipid-metabolizing activities. GDE4 and GDE7 (also known as glycerophosphodiester phosphodiesterase domain-containing proteins GDPD1 and GDPD3, respectively) are membrane-bound lyso-PLD-type enzymes which catalyze LPA generation using the same reaction as ATX[14–17]. Recombinant mouse GDE7 (mGDE7), (but not mouse GDE4, mGDE4), produces cPA in a cell-free system[17]. GDE4 and GDE7 localize to the endoplasmic reticulum (ER) and require $Mg^{2+}$ and $Ca^{2+}$ for their enzymatic activity, respectively[14,16]. However, it is unclear whether GDE7 produces cPA in living cells, and how and where GDE7 is activated by $Ca^{2+}$. It is also unknown whether intracellularly generated cPA works as a lipid mediator. This study examined the cPA-producing activity of human GDE7 (hGDE7) in living cells as well as in a cell-free system. The topology of hGDE7 on the ER membrane was determined, and the effect of GDE7 on the PPARγ pathway was analyzed in human mammary MCF-7 and mouse preadipocyte 3T3-L1 cells. The results suggest that GDE7 is involved in the production of cPA which functions as an intracellular lipid mediator.

## Results

### GDE7 produces cPA in a cell-free system.
We examined whether hGDE7 produces cPA in the presence of $Ca^{2+}$ using a cell-free system. For this purpose, FLAG-tagged hGDE7 as well as mGDE7 were stably overexpressed in COS-7 cells, which are commonly used for protein expression, and the membrane fractions were prepared. Activity of recombinant GDE7 enzymes was confirmed using a selective assay system with a fluorescent substrate, FS-3[18]. The FS-3-degrading activities of hGDE7- and mGDE7-expressing cells ($0.65 \pm 0.03$ and $1.00 \pm 0.06\ \mu mol\ h^{-1}$ per mg protein, respectively) significantly increased compared with control cells ($0.05 \pm 0.02\ \mu mol\ h^{-1}$ per mg protein) (Fig. 1a). Enzyme overexpression was confirmed by immunoblot analyses with anti-FLAG and anti-hGDE7 antibodies (Fig. 1b). Anti-FLAG antibody gave strong bands for mGDE7-expressing cells, while anti-hGDE7 antibody only presented faint bands for the same

cells; this may reflect the fact that this antibody was raised against hGDE7. Recombinant hGDE7-containing membrane fractions generated cPA in addition to LPA following the addition of [$^{14}$C] LPC substrate (Fig. 1c). The cPA-producing activity of hGDE7 ($1.81 \pm 0.58\ \mu mol\ h^{-1}$ per mg protein) was 3.7-fold higher than the LPA-producing activity ($0.50 \pm 0.17\ \mu mol\ h^{-1}$ per mg protein) (Fig. 1d). Recombinant mGDE7 also preferred to generate cPA rather than LPA, consistent with previous reports[17]. The cPA- and LPA-producing activities were $5.34 \pm 1.98$ and $0.83 \pm 0.21\ \mu mol\ h^{-1}$ per mg protein, respectively. To investigate whether cPA products are formed by GDE7 activity or through other pathways, we tested cPA-producing activity of hGDE7 in the presence of the PLD1/2 inhibitors FIPI[19] and BML 279[20] as well as the ATX inhibitor S32826[21]. The results showed that all of these inhibitors did not reduce cPA production by hGDE7 (Supplementary Fig. 1). In contrast, BrP-LPA, which inhibits ATX[22] and the FS-3-degrading activity of GDE7[18], reduced cPA production by approximately 20%. These results suggest that GDE7 and PLD2/ATX act separately. To examine $Ca^{2+}$-dependency of the cPA-producing activity of GDE7, we next investigated the GDE7 activity with or without $Ca^{2+}$, as well as in the presence of the calcium chelator EGTA. Similar to our previous results for the LPA-generating activity[16], cPA-generating activity of GDE7 was also $Ca^{2+}$-dependent (Supplementary Fig. 1). Next, in order to analyze endogenous GDE7, we also prepared the membrane fractions of wild-type human breast cancer MCF-7 cells (WT) endogenously expressing GDE7, as well as GDE7-knockout MCF-7 cells (KO)[18]. FS-3-degrading activity for KO was reduced to 37% of that for WT, and ectopic expression of mGDE7 in KO restored the activity to 242% (Fig. 1e). The loss of GDE7 in KO was confirmed by western blotting using an anti-GDE7 antibody, while mGDE7 overexpression was confirmed with an anti-FLAG antibody (Fig. 1f). cPA and LPA production rates of the KO membrane fractions were 16.7% and 48.2% of those of the WT, respectively. mGDE7 overexpression restored these rates to 292% and 434%, respectively (Fig. 1g, h). These results indicate that recombinant and endogenous proteins of hGDE7 produce not only LPA but also cPA in the cell-free system.

### GDE7 produces cPA in living cells.
We next examined whether hGDE7 produces cPA along with LPA in cells. Mass spectrometry revealed that hGDE7 overexpression in COS-7 cells produced a 2.2-fold increase in intracellular LPA levels compared to control cells (Fig. 2a), which correlated with our previous report[16]. The most abundant LPA molecular species in hGDE7-overexpressing cells was C18:1, followed by C16:0, C16:1, C18:0, and C18:2 (Fig. 2a). Polyunsaturated fatty acid-containing LPA was below the detection limit except for C18:2, even with hGDE7 overexpression. We could not detect any cPA molecular species in control COS-7 cells, while C18:0, C18:1, and C16:0 species were found in hGDE7-overexpressing cells with a total cPA amount of $7.5 \pm 2.1$ pmol per mg protein (Fig. 2b). The intracellular levels of LPA and cPA in WT were up to $91.2 \pm 23.5$ and $26.9 \pm 6.9$ pmol per mg protein, respectively. Their levels significantly decreased to 12.3% and 26.3%, respectively, in KO (Fig. 2c, d). Furthermore, mGDE7 overexpression in KO resulted in 2.3-fold and 1.4-fold increases in the LPA and cPA levels compared with the WT, respectively. Similar results were obtained with each molecular species of LPA and cPA. These results suggest that recombinant and endogenous proteins of hGDE7 produce cPA along with LPA in living cells.

### The active site of hGDE7 is directed toward the ER lumen.
We previously reported that overexpressed GDE7 localizes to the ER

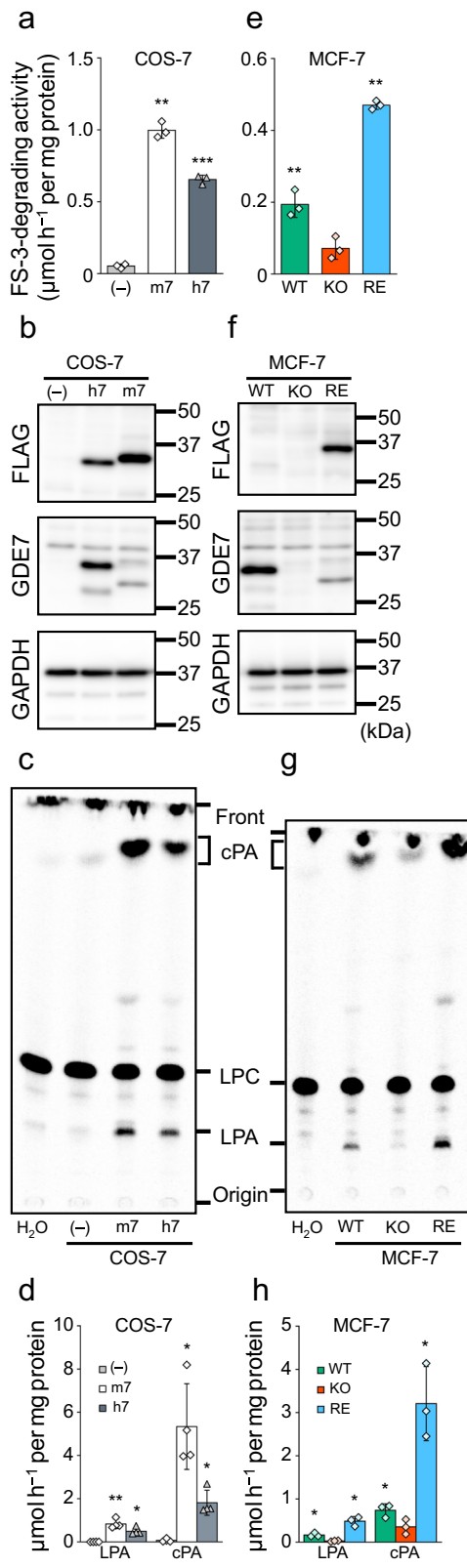

**Fig. 1 cPA-producing activity of human GDE7 in cell-free systems.** The membrane fractions of COS-7 cells (left graphs and panels) overexpressing mGDE7 (m7) and hGDE7 (h7) and control cells (−) were prepared. The membrane fractions were also prepared from wild-type (WT) cells and GDE7-deficient (KO) MCF-7 cells (right graphs and panels) and KO cells overexpressing mGDE7 (RE). **a**, **e** The membrane fractions were incubated with 5 μM FS-3 for 3 h at 37 °C, and FS-3-degrading activities were presented. Bars represent mean values ± S.D. ($n = 3$). **b**, **f** Equal amounts of membrane fraction proteins were analyzed by immunoblotting with anti-FLAG- and anti-GDE7 antibodies. Anti-GAPDH antibody was used as a loading control. Different blots were used for each antibody. **c**, **d**, **g**, **h** The membrane fractions were incubated with 25 μM 1-[$^{14}$C]oleoyl LPC for 30 min at 37 °C. Radiolabeled lipids were then extracted and separated by TLC (**c**). The positions of the origin, LPA, LPC, cPA, and the solvent front on the TLC plates are indicated. LPA- and cPA-producing activities are shown in **d** and **h** (mean values ± S.D., $n = 3$ or 4). Dunnett's test was used for analysis. *$P < 0.05$, **$P < 0.01$ (COS-7, vs. (−); MCF-7, vs. KO). All the experiments were repeated at least twice.

AlphaFold[26] (Fig. 3b & Supplementary Fig. 2). The substrate and $Ca^{2+}$ binding sites were then predicted using Firestar[27], a ligand-binding site prediction software (Fig. 3c). E71, D73, and E150 of hGDE7 were predicted as $Ca^{2+}$ binding sites, while H44, R45, E71, D73, H86, E150, K152, E180, and W277 were predicted as binding sites for glycerol 3-phosphate, a component of LPC, cPA, and LPA (Fig. 3c). Most of these putative binding sites are between the predicted TM1 and TM2. The topology of hGDE7 on the ER membrane was evaluated using the proteinase K (proK) protection assay of particulate fractions from COS-7 cells overexpressing N- and C-terminally FLAG-tagged hGDE7 (GDE7-N and GDE7-C, respectively) and analyzed by western blotting using anti-FLAG and anti-GDE7 antibodies. The proK treatment without any detergent should degrade the cytoplasmic region of the protein but not the luminal or TM region of the protein. Meanwhile, proK treatment with the zwitterionic detergent CHAPS solubilizes the ER membrane and degrades all protein regions. This assay was validated based on the observed degradation of ER lumen localized protein disulfide isomerase (PDI) (around 57 kDa) following proK and CHAPS treatment, but not by the proK treatment alone (Fig. 4a). ProK treatment of the GDE7-N particulate fraction without detergent produced an immunoreactive band with a slightly lower molecular mass using anti-FLAG antibody (Fig. 4a). This suggests that the N-terminus of hGDE7 is located in the ER lumen. In contrast, the protein band of GDE7-C was lost by a similar proK treatment, suggesting that the C-terminus of hGDE7 localizes to the cytoplasmic side (Fig. 4a). Immunoblot analyses of proK-treated GDE7-N and GDE7-C particulate fractions with anti-GDE7 antibody indicated a protein band around 32 kDa, suggesting that the immunogenic sequence (aa 107–177) localizes to the ER luminal side (Fig. 4a). Furthermore, similar results were obtained with endogenous GDE7 in MCF-7 cells using anti-GDE7 antibody (Fig. 4b). These results indicate that the N-terminus of hGDE7 localizes to the ER luminal side, the C-terminus is on the cytoplasmic side, and the active site of hGDE7 is directed toward the ER lumen (Fig. 4c).

GDE7 topology was also examined by immunocytochemistry of COS-7 cells overexpressing GDE7-N and GDE7-C. A low digitonin concentration selectively permeabilizes the plasma membrane, whereas Triton X-100 permeabilizes all cellular membranes[28]. Thus, cytoplasmic antigens should be stained when permeabilized with either digitonin or Triton X-100, while antigens in the ER lumen are stained only when permeabilized with Triton X-100. We confirmed that PDI proteins in the ER

and its activity is stimulated by $Ca^{2+}$[16]. Analysis of hGDE7 topology on the ER membrane using three prediction programs (HMMTOP[23], TMHMM[24], and SPLIT[25]) suggested that it has up to three predicted transmembrane (TM) domains (around amino acid residues 3–23 (TM1), 200–219 (TM2), and 240–259 (TM3)) (Fig. 3a). A 3D structure model of hGDE7 was created using

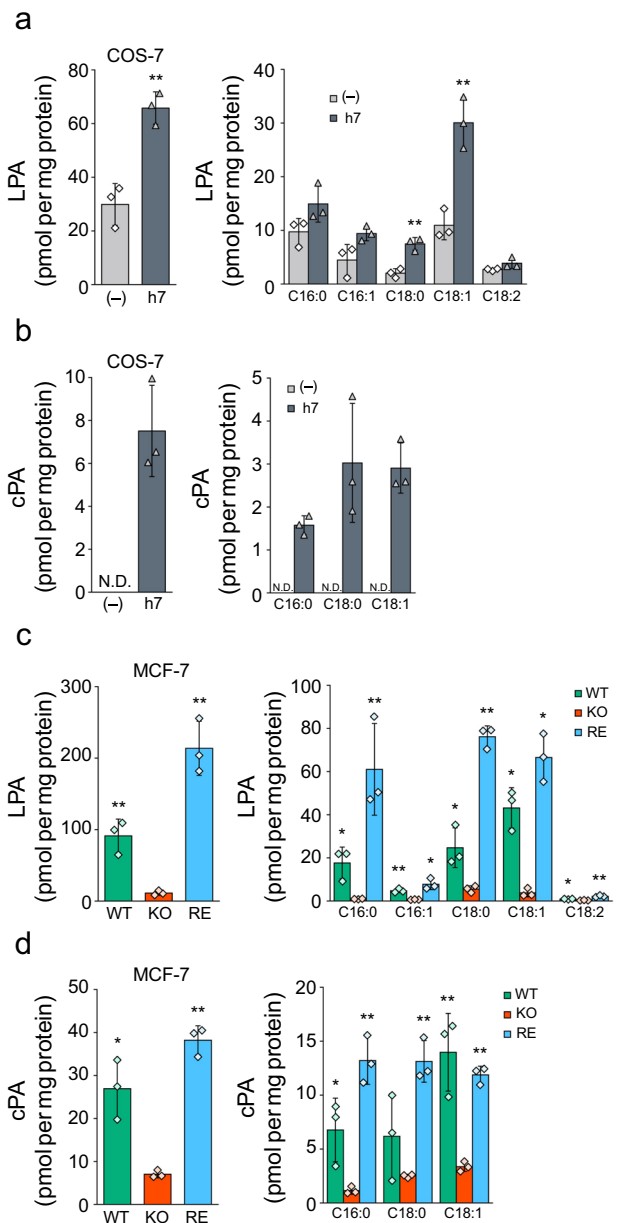

**Fig. 2 cPA-producing activity of human GDE7 in cells.** The levels of LPA (**a**) and cPA (**b**) in control (–) and hGDE7-overexpressing (h7) COS-7 cells analyzed by LC-MS/MS. The levels of LPA (**c**) and cPA (**d**) analyzed with wild-type (WT) and GDE7-deficient (KO) MCF-7 cells and KO cells overexpressing mGDE7 (RE). The graphs present the total levels (left) and the levels of each molecular species (right). Bars represent mean values ± S.D (n = 3). Log transformed data were used for Dunnett's test analysis. *P < 0.05, **P < 0.01 (COS-7, vs. (–); MCF-7, vs. KO). N.D., not detected.

lumen were immunodetected only when permeabilized with Triton X-100 and that cytoplasmic tubulin β3 proteins were detected following treatment with either digitonin or Triton X-100 (Fig. 4d). GDE7-N was detected using an anti-FLAG antibody only when permeabilized with Triton X-100, while GDE7-C was detected when permeabilized with either digitonin or Triton X-100 (Fig. 4d). These results support the proK protection assays indicating that the N- and C-terminus of hGDE7 are located on the ER lumen and cytoplasm, respectively (Fig. 4e). Together, these observations indicate that the hGDE7 active site is directed toward the ER lumen (Fig. 4f).

**Mutagenesis reveals the importance of F227 and Y238 for hGDE7 catalytic activity.** The Smith-Waterman algorithm[29] revealed that the amino acid sequences of hGDE4 and hGDE7 are 42.1% identical and 64.1% similar (Fig. 5a). Since mGDE4 produces LPA but not cPA in cell-free assays[17], we compared the amino acid sequences of GDE4 and GDE7 from humans and mice to determine which amino acid residues in hGDE7 are important for its ability to produce cPA. We considered that hydrophobic amino acid residues that prevent water molecules from approaching the active pocket are important for cPA production by transphosphatidylation. Therefore, F227E and Y238K substitutions in hGDE7 were selected based on the following three criteria: (i) hydrophobic amino acid residues in hGDE7 are replaced with hydrophilic amino acid residues in hGDE4, (ii) amino acid residues are located on the surface of the active pocket of the structural model, and (iii) amino acid residues are conserved between hGDE7 and mGDE7. The hGDE7 mutants F227E, Y238K, and F227E/Y238K were then expressed in COS-7 cells. The expression level of Y238K was higher compared to wild-type hGDE7, while F227E and F227E/Y238K had lower expression than the wild-type according to western blotting with anti-FLAG and anti-GDE7 antibodies using the same amount (10 μg) of the membrane fractions (Supplementary Fig. 3). We used the amounts of membrane fractions giving similar band intensities of hGDE7 by western blotting in the following enzyme assays (Fig. 5b). Membrane fractions from control cells were added to retain the same amounts of endogenous COS-7 proteins (10 μg). The FS-3-degrading activity of F227E (0.071 ± 0.005 μmol h$^{-1}$ per mg protein) was significantly lower than that of the wild-type (0.136 ± 0.002 μmol h$^{-1}$ per mg protein). The activity of Y238K (0.042 ± 0.004 μmol h$^{-1}$ per mg protein) was almost the same as the control (0.035 ± 0.008 μmol h$^{-1}$ per mg protein), while that of F227E/Y238K (0.028 ± 0.003 μmol h$^{-1}$ per mg protein) was below the basal level (Fig. 5c). The LPA- and cPA-producing activities from [$^{14}$C]LPC using 0.5 μg of protein sample (Fig. 5d, e) showed a similar tendency to those of FS-3-degrading activity, although the decrease in cPA-producing activity of F227E was not statistically significant. These mutation assays did not allow us to determine the critical amino acid residues of hGDE7 for cPA production; however, both F227 and Y238 are critical amino acid residues for the catalytic activity of hGDE7.

**cPA produced by GDE7 suppresses PPARγ target genes.** It has been reported that cPA produced by PLD2 functions as an antagonist of PPARγ[8]. Therefore, we hypothesized that if cPA produced by GDE7 functions as a lipid mediator, its PPARγ-suppressive action would also overlap with that of PLD2. Therefore, we examined the mRNA expression levels of PPARγ downstream factors, CD36 and CYP27A1, in MCF-7 cells. Both of these levels were derepressed in KO compared to WT, and their levels were significantly suppressed by the expression of mGDE7 in KO (Fig. 6a). The expression levels of PPARG were similar between three types of MCF-7 cells. Since PPARγ signaling is crucial for adipogenic differentiation[30], we next analyzed mouse 3T3-L1 pre-adipocytes as a clue to understand physiological relevance of cPA produced by GDE7. We then overexpressed hGDE7 in the cells after the activation of PPARγ by rosiglitazone (ROSI) treatment for 20 h in serum-free medium, and confirmed the overexpression by GDPD3 (exogenous hGDE7 mRNA) levels (Fig. 6b). The overexpression of hGDE7 significantly suppressed mRNA expression of PPARγ downstream factors, Cyp27a1, Adipoq, and Fabp4. Another PPARγ downstream factor, Cd36, was also suppressed, although the difference did not reach statistical significance. The expression levels of Gdpd3 (endogenous mGDE7 mRNA) and Pparg were not significantly changed by the overexpression or ROSI treatment. These results suggest that cPA

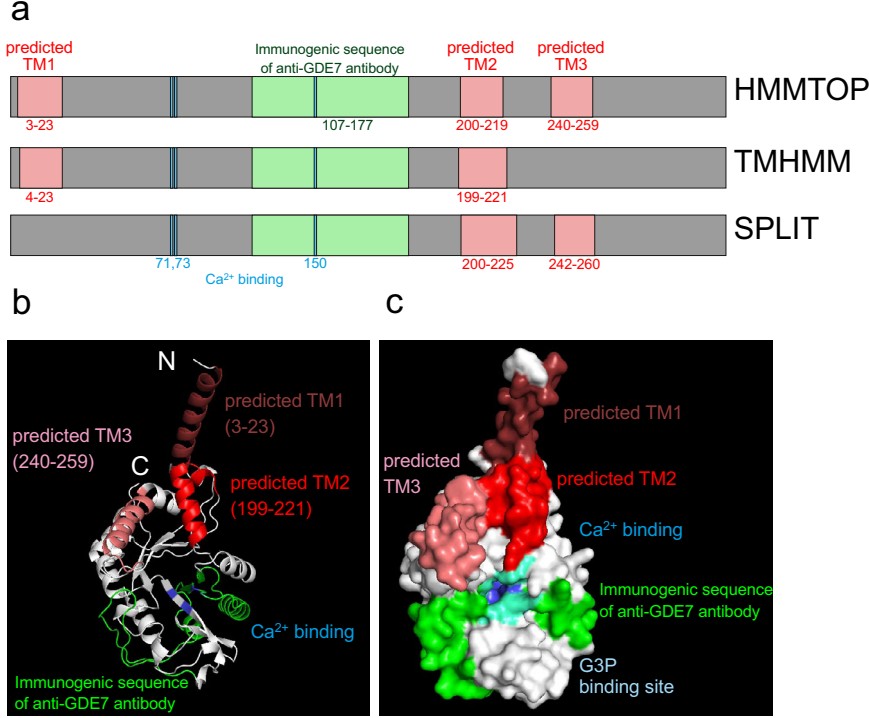

**Fig. 3 Prediction of TM domains and active site of hGDE7. a** The estimated TM domains, $Ca^{2+}$ binding site, and immunogenic sequence of anti-GDE7 antibody were determined with the indicated TM domain prediction software. Numbers indicate the positions of amino acid residues. **b, c** The predicted conformation of hGDE7 is shown as a cartoon (**b**) or surface (**c**) representation. The estimated TM domains, immunogenic sequence of the anti-GDE7 antibody, $Ca^{2+}$ binding site, and glycerol 3-phosphate (G3P) binding site are shown.

produced by GDE7 in the ER lumen is an intracellular lipid mediator. On the other hand, we could not detect the protein expressions of these PPARγ downstream factors under the same conditions. We then treated 3T3-L1 cells with ROSI in the presence of 10% calf serum for 1 week and could detect the increased protein expressions (Fig. 6c). However, overexpression of hGDE7 hardly suppressed the expressions. This could be because ROSI is a fairly powerful driver of adipogenic differentiation. It also could be due to the influence of other lipid mediators in the calf serum. Furthermore, pre-adipocytes are not as multipotential so it might be harder to show a suppression of adipogenic differentiation by hGDE7 in these cells. It is possible that with other types of adipogenic media, a different result could be obtained.

## Discussion

This study demonstrated that hGDE7 has intracellular cPA-producing properties and the active site of hGDE7 is directed toward the lumen of the ER. Moreover, GDE7 suppressed the PPARγ pathway. These results suggest that intracellular cPA produced by GDE7 localized in the ER functions as a lipid mediator (Fig. 7).

ATX and PLD2 are involved in the production of cPA and LPA from LPC[8,12]. Moreover, mouse GDE7 catalyzes these reactions in a cell-free system[17]. Consistent with these findings, we confirmed that hGDE7 produces cPA and LPA in cells as well as in a cell-free system. ATX preferentially generates LPA rather than cPA under near-physiological conditions in a cell-free system, with a 1:4 ratio of cPA:LPA production[12]. In contrast, we found that the ratio of cPA:LPA production by hGDE7 was approximately 4:1 to 5:1 in a cell-free system (Fig. 1). However, our assay system contains 5% ethanol, and this ratio should be examined under conditions similar to biological membranes. In contrast, the apparent ratio of cPA:LPA production by hGDE7 was 1:3 to 1:9 in cells (Fig. 2). This difference may result from the

involvement of other LPA-producing enzymes such as GDE4 and acylglycerol kinase[31] since MCF-7 cells have poor ATX expression[32]. It is also possible that cPA was hydrolyzed to LPA non-enzymatically in the presence of water[33]. Alternatively, it is possible that cPA produced by GDE7 is converted to β-LPA since it has been reported that cPA produced by ATX is converted to β-LPA[34]. In humans, GDE7 is abundant in the kidney, prostate, ovary, and placenta[16], while PLD2 is widely expressed in various tissues[35]. Thus, ATX, GDE7, and PLD2 produce cPA but differ in their sites of action, substrate specificity, and tissue distribution: ATX is mainly involved in blood cPA production, while GDE7 and PLD2 produce cPA intracellularly, and each enzyme may have a unique role. Due to the limitation of low human blood levels of cPA, its quantification in human samples is difficult and has not been extensively studied; however, it is possible that the high local concentrations of cPA could exert physiological significance. Recently, methods for the quantification of lysophospholipids including cPA have been established[36], and it is expected that the role of cPA in health and diseases will be clarified in the near future. The LPC in plasma is involved in cell migration, cytokine production, oxidative stress, and apoptosis[37], while intracellular LPC is involved in ER stress[38]. Therefore, GDE7 may be involved in LPC removal for ER homeostasis, in addition to the production of LPA and cPA as lipid mediators.

GDE family members GDE1 and GDE4 have two TM domains and their active sites are on the luminal side[39,40]. Similarly, the present study revealed that the active site of GDE7 is on the luminal side of the ER membrane based on topological predictions, the proK protection assay, and immunocytochemistry (Figs. 3 and 4). GDE7 activity is $Ca^{2+}$-dependent[16]; therefore, GDE7 should be activated by the abundant levels of $Ca^{2+}$ in the ER lumen to constitutively produce LPA and cPA. In contrast, PLD2 localizes to the plasma membrane and generates cPA in a stimulus-coupled manner[8]. Thus, GDE7 and PLD2 may have

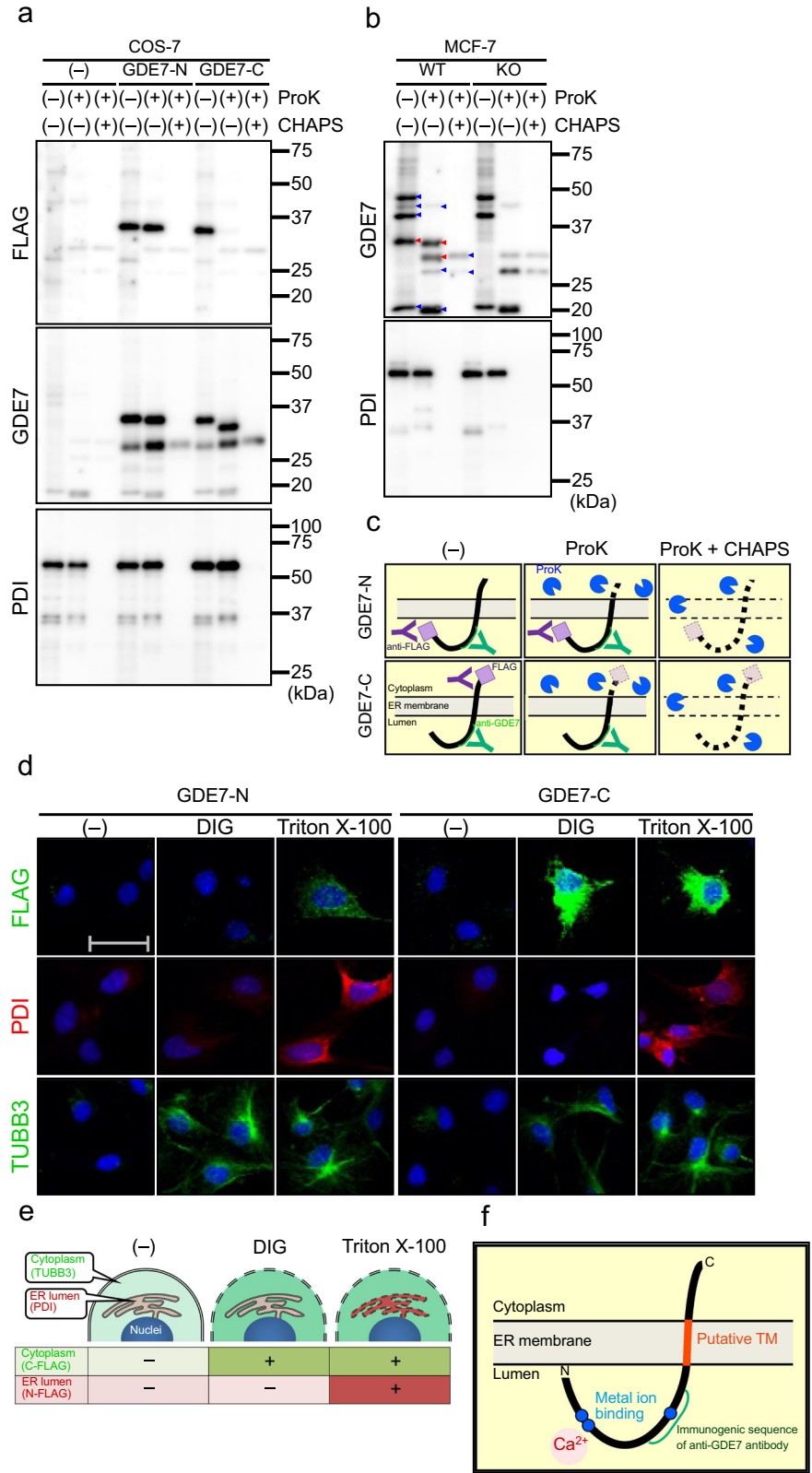

distinct roles in terms of localization and stimulus responsiveness. The N-terminally FLAG-tagged GDE7 used in this study contains a FLAG region followed by a 5×Gly linker which is only two or three amino acid residues from the presumed TM1. AlphaFold predicts that adding a FLAG region to the N- or C-terminus of hGDE7 has no significant effect on the structural conformation. However, the negative charge of FLAG might alter translocon-

mediated protein translocation and TM domain integration. In the present study, the intracellular localization of GDE7 was analyzed using cells overexpressing FLAG-tagged hGDE7. Thus, the localization of endogenous GDE7 remains unknown due to the unavailability of reliable antibodies applicable for immunocytochemistry. Further studies with more accurate structural information are required in the future.

**Fig. 4 hGDE7 Topology. a** Equal amounts of particulate fraction proteins from COS-7 cells overexpressing N- or C-terminally FLAG-tagged hGDE7 (GDE7-N and GDE7-C, respectively) or the control cells (–) were treated with proK and CHAPS as indicated, and analyzed by immunoblotting using anti-FLAG-, anti-GDE7-, and anti-PDI antibodies. Different blots were used for each antibody. **b** Equal amounts of particulate fraction proteins from wild-type (WT) and GDE7-deficient (KO) MCF-7 cells were treated with proK, and CHAPS as indicated, and analyzed by immunoblotting using anti-GDE7- and anti-PDI antibodies. Red and blue arrowheads show GDE7-specific and nonspecific bands, respectively. Different blots were used for each antibody. **c** Schematic illustrations of the proK protection assay. **d** Representative immunofluorescence images of COS-7 cells overexpressing GDE7-N or GDE7-C using anti-FLAG-, anti-PDI-, and anti-TUBB3 antibodies. The cells were permeabilized with digitonin (DIG), Triton X-100, or without them (–). The scale bar represents 50 μm. **e** Schematic illustration of the permeabilization with digitonin and Triton X-100. **f** Schematic illustration of hGDE7 topology. All the experiments were repeated at least twice.

Although the active sites of GDE4 and GDE7 are highly conserved, their cPA-producing activities are quite different[17]. We constructed GDE7 mutants by replacing the hydrophobic amino acid residues in the active site (F227 and Y238) with hGDE4-like hydrophilic amino acid residues to investigate whether they are important for cPA-producing activity of hGDE7 (Fig. 5). LPA-producing activity was preferentially decreased in F227E, while both LPA- and cPA-producing activities were reduced in Y238K. Furthermore, both activities were completely lost in F227E/Y238K. Previous studies of *Streptomyces* PLD depicted that amino acid residues forming the active site entrance are involved in substrate recognition, while those forming the active site stabilize the conformation[41]. It was predicted that F227 forms the active site and Y238 forms the entrance of hGDE7 (Supplementary Fig. 2). Thus, F227 and Y238 might be important for stability and substrate recognition, respectively. Although F227 and Y238 are important for hGDE7 activity, further structural biological studies are needed to clarify how GDE7 produces cPA.

Currently, no specific targets for cPA have been reported. Extracellularly produced cPA may act on LPA receptors such as LPA1–3[42], LPA5[43], and LPA6/P2Y5[44,45]. However, it is unclear whether cPA acts in a similar or opposite manner to LPA. At least for PPARγ, LPA works as an agonist[7] and cPA as an antagonist[8]. cPA intracellularly produced by PLD2 suppresses PPARγ[8] and inhibits intimal thickening and proliferation of colon cancer cells by inducing apoptosis[46]. The present study similarly showed that GDE7 suppressed the expression of PPARγ target genes (Fig. 6). Thus, cPA was suggested as an intracellular lipid mediator, and it is necessary to clarify whether and how intracellular cPA produced by GDE7 contributes to the pathogenesis via the PPARγ pathway. Furthermore, the possible involvement of another intracellular cPA target, adenine nucleotide translocase, which interacts with 2-carba-cPA[47] should be considered.

It has been reported that activation of PPARγ could inhibit growth and progression of malignant breast cancer[48]. Thus, higher expression of GDE7 could suppress PPARγ via increased cPA levels and potentially accelerate tumor progression. The present study also showed that cPA produced by GDE7 suppresses PPARγ pathway in 3T3-L1 cells, suggesting a relationship with adipogenic differentiation. Recently, Shimizu et al. showed that mRNA expression level of GDE7 was increased under hypoxic conditions in Caco-2 cells[49]. Since the endogenous mRNA level of GDE7 was found to be low in 3T3-L1 cells in the present study, it is required to analyze GDE7 activators and transcription factors that regulate its expression in the future. Further studies using primary bone marrow MSCs are also needed to determine whether the suppression of PPARγ by cPA produced by GDE7 is involved in the signaling that determines MSC differentiation into adipocytes[30].

GDE7 is involved in cancer stem cell survival[50], hepatic lipidosis[51], and noise-induced hearing loss[52]. Furthermore, *GDPD3* encoding hGDE7 is located at chromosome 16p11.2, and copy number variation in this region is associated with increased risk of neuropsychiatric diseases[53] and obesity-related syndromes[54].

However, it is unclear whether and how the lipid mediators intracellularly produced by GDE7 contribute to these pathologies. Protein carriers of lipid mediators are essential for the regulation of physiological functions and pathological processes[11]. LPA and cPA bind to albumin in the blood[55–57] and act as lipid mediators. Meanwhile, intracellularly produced LPA functions as a PPARγ agonist[58], and fatty acid binding protein 3 is an LPA carrier protein in PPARγ activation[59]. Further studies are required to clarify the intracellular cPA-specific binding proteins and the transport mechanisms from the ER lumen to demonstrate that intracellular cPA also plays an important role in vivo. Future studies should focus on not only LPA but also cPA as the intracellular products of GDE7 when analyzing its association with various pathological conditions.

In conclusion, we demonstrated that GDE7 produces cPA in the ER lumen, which could function as an intracellular lipid mediator. These findings lead to a better understanding of the biological role of GDE7 as well as its products, LPA and cPA.

## Methods

**Materials**. FS-3 and BrP-LPA were purchased from Echelon Biosciences (Salt Lake City, UT). Anti-FLAG M2 (F1804) and anti-PDI (P7372) antibodies as well as FIPI were purchased from Sigma-Aldrich (St. Louis, MO), anti-GDE7 antibody (HPA041148) was from Atlas Antibodies (Stockholm, Sweden), anti-GAPDH antibody (M171-3) was from MBL (Tokyo, Japan), anti-tubulin β-3 antibody (921001) was from BioLegend (San Diego, CA). Anti-PPARγ (#2435), anti-adiponectin (#2789), anti-CD36 (#28109), and anti-FABP4 (#2120) antibodies were from Cell Signaling Technology (Danvers, MA). [14C]LPC (1-[14C]oleoyl-*sn*-glycerophosphocholine) was purchased from American Radiolabeled Chemicals (ARC 3094; St. Louis, MO). LPC (1-oleoyl-*sn*-glycerophosphocholine) and cPA (1-oleoyl-*sn*-glycero-2,3-cyclic-phosphate) were from Avanti Polar Lipids (Alabaster, AL). Nonidet P-40 and proK were procured from Nacalai Tesque (Kyoto, Japan). CHAPS was purchased from Dojindo (Kumamoto, Japan). BML 279 was purchased from Abcam (Cambridge, UK). S32826 was procured from Cayman Chemical (Ann Arbor, MI).

**Cell line**. Transformed African green monkey kidney fibroblast COS-7 cells (RCB0539, Riken BioResource Research Center, Tsukuba, Japan) were maintained in Dulbecco's modified Eagle's medium (DMEM) (FUJIFILM Wako, Osaka, Japan) supplemented with 10% (v/v) fetal bovine serum (FBS) (Biofluids, Fleming Island, FL) and nonessential amino acids (FUJIFILM Wako). Human breast cancer MCF-7 cells (Health Science Research Resources Bank, Osaka, Japan) were maintained in DMEM supplemented with 15% (v/v) FBS. A knockout cell line for human GDE7 gene (*GDPD3*) was previously established[18]. 3T3-L1 cells (Japanese Collection of Research Bioresources Cell Bank, Ibaraki, Japan) were maintained in DMEM (1 g L$^{-1}$ glucose) supplemented with 10% (v/v) calf serum (Cytiva, Marlborough, MA). These cell lines were cultured at 37 °C in a humidified atmosphere of 5% $CO_2$ and 95% air.

**Plasmid construction and lentiviral preparation**. cDNA of human or mouse GDE7 with a C-terminal FLAG tag was previously constructed[16]. A cDNA sequence encoding a FLAG peptide was introduced by PCR using PrimeSTAR HS DNA Polymerase (TaKaRa Bio, Kusatsu, Japan) to construct cDNA for N-terminally FLAG-tagged GDE7. This was generated using a forward primer containing a XbaI site, Kozak sequence, FLAG, and 5×Gly linker, and a reverse primer with a BamHI site. Site-directed mutagenesis was performed by overlap extension PCR. Each DNA fragment was amplified using primer sets shown in Supplementary Table 1 and subcloned into the third generation lentiviral backbone vector containing a hygromycin resistant gene as previously described[60].

COS-7, MCF-7, and 3T3-L1 cells were transduced with each obtained lentivirus using 8 μg mL$^{-1}$ hexadimethrine bromide (Sigma-Aldrich) and then selected with hygromycin B (Nacalai Tesque).

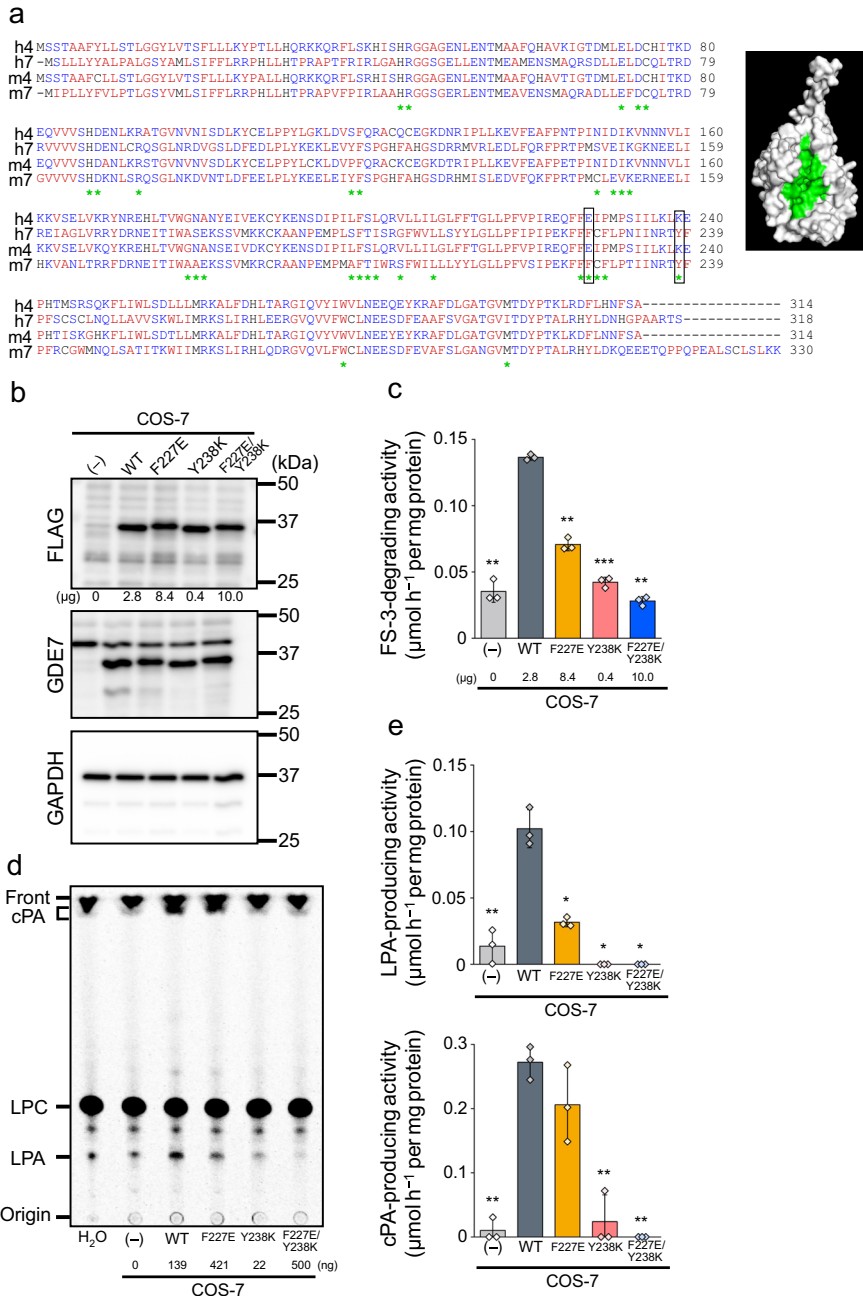

**Fig. 5 hGDE7 mutations. a** Amino acid sequences of hGDE4 (h4), hGDE7 (h7), mGDE4 (m4), and mGDE7 (m7). Sequences were aligned using the Smith-Waterman algorithm. Blue and red letters indicate hydrophilic and hydrophobic amino acids, respectively. Green asterisks indicate amino acids on the surface of the predicted GDE7 active pocket. Open boxes indicate amino acids used for mutations (F227 and Y238). **b–e** Analysis of membrane fractions from COS-7 cells overexpressing GDE7 wild-type (WT), each GDE7 mutant, and control COS-7 cells (–). **b** The membrane fractions were subjected to immunoblotting with anti-FLAG- and anti-GDE7 antibodies. Anti-GAPDH antibody was used as a loading control. Different blots were used for each antibody. **c** The membrane fractions of COS-7 cells overexpressing each mutant GDE7 were incubated with FS-3, and FS-3-degrading activities were shown. Bars represent mean values ± S.D. ($n = 3$). **d** and **e** The membrane fractions were incubated with 1-[$^{14}$C]oleoyl LPC. Radiolabeled lipids were then extracted and separated by TLC (**d**). The positions of the origin, LPA, LPC, cPA, and the solvent front on the TLC plate are indicated. **e** LPA- and cPA-producing activities (mean values ± S.D., $n = 3$). The amounts of membrane fractions prepared from GDE7-expressing cells are shown in **b–d**. For each experiment, the total amounts were adjusted to 10 μg (**a** and **c**) or 500 ng (**d**) by adding the membrane fraction from the control cells. Dunnett's test was conducted for analysis. *$P < 0.05$, **$P < 0.01$, ***$P < 0.001$ (vs. WT). All the experiments were repeated at least twice.

**Immunoblot analysis**. Immunoblot analysis was performed as previously described[18,61] with modifications. COS-7 cells were sonicated three times for 3 s each in 50 mM Tris–HCl buffer (pH 7.4) and were then ultracentrifuged at 105,000 × g for 1 h at 4 °C. The resultant pellets were suspended in 20 mM Tris–HCl buffer (pH 7.4) and used as membrane fractions. 3T3-L1 cells were lysed in lysis buffer (50 mM Tris–HCl (pH 7.5), 150 mM NaCl, 2 mM EDTA, 1% Nonidet P-40, and 0.1% sodium dodecyl sulfate (SDS)) with Halt Protease and Phosphatase Inhibitor Cocktail

(Thermo Fisher Scientific, Waltham, MA) and were centrifuged at 12,000 × g for 15 min at 4 °C. The supernatants were then used as cell lysates. The membrane fractions and cell lysates were incubated at 95 °C for 5 min in SDS sample buffer (60 mM Tris–HCl (pH 6.8), 2% SDS, 10% glycerol, 100 mM dithiothreitol, and 0.01% bromophenol blue). The obtained samples were separated on SDS-polyacrylamide gel and electrotransferred onto polyvinylidene difluoride (PVDF) membranes (Immobilon-E, Merck Millipore, MA). The PVDF membranes were blocked with

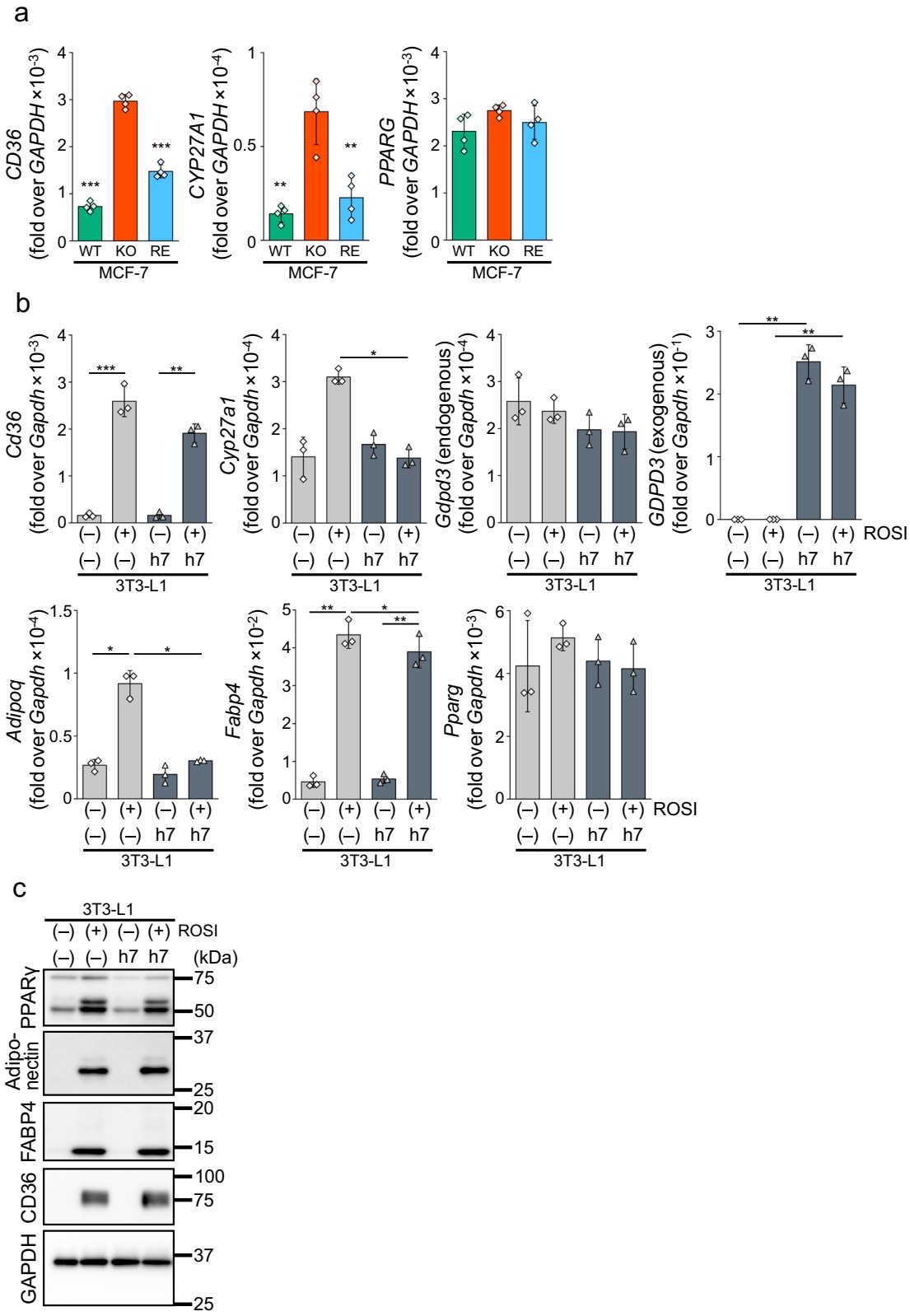

phosphate-buffered saline (PBS) containing 0.1% Tween 20 (PBS-T) and 5% skim milk (Morinaga Milk Industry, Tokyo, Japan) for 1 h at 25 °C. They were then probed with anti-FLAG (1:1000 dilution in 2% BSA/PBS-T), anti-GDE7 (1:1000), anti-GAPDH (1:1000), anti-PDI (1:1000), anti-PPARγ (1:1000), anti-adiponectin (1:1000), anti-FABP4 (1:1000), and anti-CD36 antibodies (1:1000) for 18 h at 4 °C. After the membranes were washed three times with PBS-T, the bound antibodies were visualized using horseradish peroxidase-linked anti-rabbit or anti-mouse IgG secondary antibody (Cell Signaling Technology, 1:1000) depending on the primary

antibody. Protein expression signals were detected with Pierce ECL western blotting substrate (Thermo Fisher Scientific), and the chemiluminescent signals were quantified using an Amersham Imager 680 (Cytiva).

**Enzyme assay**. Fluorescence-based enzyme assays were based on a previous method[18]. Briefly, membrane fractions (5 μg of protein) were incubated with 5 μM FS-3 for 3 h at 37 °C in 60 μL of 50 mM Tris–HCl buffer (pH 7.4) in the presence of 2 mM $CaCl_2$ and 0.1% (w/v) Nonidet P-40.

**Fig. 6 Suppression of the PPARγ pathway by intracellular cPA. a** Wild-type (WT), GDE7-deficient (KO) MCF-7 cells, and KO cells overexpressing mGDE7 (RE) were cultured in serum-free medium for 20 h. Total RNA was isolated and analyzed by reverse transcription qPCR for mRNA levels of *CD36*, *CYP27A1*, and *PPARG*. Values are expressed as the ratios to the *GAPDH* levels (mean values ± S.D., n = 4). Dunnett's test was conducted for analysis. **P < 0.01, ***P < 0.001 (vs. KO). **b** 3T3-L1 cells overexpressing hGDE7 (h7) and control 3T3-L1 cells (–) were cultured in serum-free medium with or without 1 μM rosiglitazone (ROSI) for 20 h. Total RNA was isolated and analyzed by reverse transcription qPCR for mRNA levels of *Cd36*, *Cyp27a1*, *Gdpd3* (endogenous GDE7), *GDPD3* (exogenous GDE7), *Adipoq*, *Fabp4*, and *Pparg*. Values are expressed as the ratios to the *Gapdh* levels (mean values ± S.D., n = 3). Tukey test was conducted for analysis. *P < 0.05, **P < 0.01, ***P < 0.001. **c** 3T3-L1 cells overexpressing hGDE7 (h7) and control 3T3-L1 cells (–) were treated with or without 1 μM ROSI in the presence of 10% calf serum for 1 week (medium change every other day). The cell lysates containing equal amounts of proteins were subjected to immunoblotting with anti-PPARγ, -adiponectin, -FABP4, and -CD36 antibodies. GAPDH was used as a loading control. Different blots were used for each antibody. All the experiments were repeated at least twice.

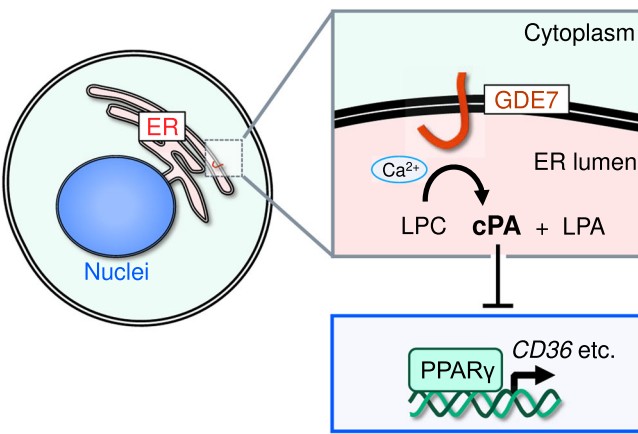

**Fig. 7 A proposed model of cPA production by GDE7.** GDE7 is activated by $Ca^{2+}$ in the ER lumen and produces cPA which inhibits the PPARγ pathway.

Radioactivity-based enzyme assays were performed as previously described[18] with modifications. The membrane fractions were incubated at 37 °C for 30 min with 25 μM of [$^{14}$C]LPC (20,000 cpm, dissolved in 5 μL of ethanol) in 100 μL of 50 mM Tris–HCl (pH 7.4) containing 2 mM of $CaCl_2$ and 100 μM of phosphatase inhibitor sodium orthovanadate to protect the product from degradation[14]. The reactions were terminated by adding 0.32 mL chloroform/methanol/1 M citric acid (8:4:1, by vol.) with 5 mM butylated hydroxyanisole as an antioxidant. After centrifugation, the organic layer was transferred to a new tube, and the aqueous layer was further extracted with 250 μL of chloroform/methanol (17:3, v/v). The combined organic layer was dried under a gentle stream of nitrogen. The obtained lipids were dissolved in chloroform/methanol (2:1) and spotted on a silica gel-coated aluminum thin layer chromatography (TLC) sheet (20-cm height; Merck, Darmstadt, Germany), and developed at 4 °C for 100 min using chloroform/methanol/28% (w/w) ammonium hydroxide (60:35:8, by vol.)[17]. The radioactivity of the substrate and product on the plate was quantified using an image analyzer (Typhoon 9400; Cytiva), and the enzyme activity was then calculated.

**Lipid analyses by liquid chromatography-tandem mass spectrometry (LC-MS/MS).** The cells were cultured to 100% confluence in a 100-mm dish, harvested by scraping, and stored at −80 °C until use. Lipid extraction was performed according to previous studies[16,62], and one hundred pmol of C17:0 LPA was mixed as an internal standard at this step. The amounts of LPA and cPA species were determined by the ratios of the peak areas in the following LC-MS/MS experiments.

LC-MS/MS was performed using a quadrupole-linear ion trap hybrid mass spectrometry system, 4000 Q TRAP (Applied Biosystems/MDS Sciex, Concord, Ontario, Canada) with an 1100 liquid chromatography system (Agilent Technologies, Wilmington, DE) combined with an HTS-PAL autosampler (CTC Analytics AG, Zwingen, Switzerland) as previously described[16] with modifications. LPA and cPA species were separated using a Tosoh TSK-ODS-100Z column (150 mm × 2 mm, 5 μm particle size) with methanol/water (95:5, v/v) containing 0.05 M ammonium formate at a flow rate of 0.20 mL min⁻¹ at 42 °C. Five microliter aliquots of the test solution were routinely applied using the autosampler. The negative ion mode of operation was used with multiple reaction monitoring, and deprotonated molecular ion and deprotonated cyclic glycerophosphate at $m/z$ 153 were selected for Q1 and Q3, respectively. Values were calculated from the ratios of the ion peak areas of the analyzing target molecules to that of the internal standard, C17:0 LPA. The ionization efficiency of cPA species was found to be higher than that of LPA; therefore, the values for cPA species were multiplied by a correction factor of 0.239. This factor was

determined by the slope of the calibration curve obtained from the plots of peak area ratios vs. molar ratios between authentic molecules, C17:0 LPA and C18:1 cPA. Log transformed data were used for statistical analyses.

**Structural prediction.** TM domains of hGDE7 were predicted by HMMTOP[23], TMHMM-2.0[24], and SPLIT4.0[25]. Ligand and calcium ion binding sites were predicted using Firestar[27]. The spatial structure of GDE7 was modeled and visualized with AlphaFold[26] and PyMOL 2.3 software.

**Topology.** ProK protection assays were performed as previously described[40] with modifications. Briefly, each cell suspension was prepared in buffer P (250 mM sucrose and 1 mM ethylenediaminetetraacetic acid (EDTA) in 50 mM Tris–HCl, pH 7.4) and homogenized by passing through a 27 G needle 20 times on ice. Nuclei and unbroken cells were removed by centrifugation at 600 × g for 5 min at 4 °C, and the supernatants were further centrifuged using an Optima MAX-TL (Beckman Coulter, Brea, CA) at 105,000 × g for 1 h at 4 °C. The resulting pellets were resuspended in buffer P and used as the particulate fractions. The particulate fraction (30 μg of protein) was incubated with 150 μg mL⁻¹ proK in the presence or absence of 1% (w/v) CHAPS for 60 min at 50 °C. The reaction was terminated by the addition of 5 mM phenylmethylsulphonyl fluoride (PMSF) (Sigma-Aldrich). Samples were immediately added to SDS sample buffer, heated at 95 °C for 10 min, and subjected to SDS-PAGE followed by immunoblotting.

Immunocytochemistry involved fixing COS-7 cells in a poly-L-lysine-coated 8-well Lab-Tek II chamber slide (Thermo Fisher Scientific) with 4% (w/v) paraformaldehyde in PBS for 10 min at 25 °C. Cells were rinsed with PBS, and then permeabilized with 0.1% (v/v) Triton X-100 (FUJIFILM Wako) in PBS or 0.001% (w/v) digitonin (Sigma-Aldrich) in PBS containing 10 mM HEPES (pH 7.5), 300 mM sucrose, 100 mM KCl, 2.5 mM $MgCl_2$, and 0.5% (w/v) BSA on ice for 15 min. After blocking with 1% (w/v) BSA/PBS for 60 min at 25 °C, the samples were incubated with specific antibodies diluted in blocking buffer (FLAG, 1:1000; PDI, 1:250; tubulin β-3, 1:1000) for 16 h at 4 °C, followed by incubation with Alexa Fluor-conjugated secondary antibodies (diluted 1:1000 with the blocking buffer) for 1 h at 25 °C and then with 10 μg mL⁻¹ Hoechst 33342 solution for 5 min to stain the nuclei. Fluorescence was then observed using a laser-scanning confocal microscopy (LSM700; Carl Zeiss, Jena, Germany).

**RNA isolation and quantitative PCR (qPCR).** MCF-7 and 3T3-L1 cells were cultured in serum-free media containing 0.2% (w/v) fatty acid-free BSA (FUJIFILM Wako) for 20 h. 3T3-L1 cells were treated with 1 μM ROSI (FUJIFILM Wako) as indicated. RNA isolation, cDNA synthesis, and qPCR were performed as previously described[63]. Briefly, RNA was isolated from the cells by using TRIzol reagent (Thermo Fisher Scientific). The quality and concentration of the isolated RNA were estimated by measuring the absorbance at 260 and 280 nm on a Nanodrop One spectrophotometer (Thermo Fisher Scientific). cDNA was then synthesized with PrimeScript RT reagent kit (TaKaRa Bio). The relative expression of gene was quantified by qPCR using a TB Green Premix Ex Taq II (TaKaRa Bio), which was performed using a StepOnePlus Real-Time PCR System (Applied Biosystems) in accordance with the manufacture's protocols. The primers used are listed in Supplementary Table 1. An average threshold cycle (Ct) value was calculated and normalized to that of housekeeping gene *GAPDH* or *Gapdh* to obtain the ΔCt value.

**Statistics and reproducibility.** All data are expressed as the mean ± SD. Statistical analyses were performed using Prism ver. 8.42 (GraphPad Software; San Diego, CA). The sample sizes from one time experiment, reproducibility, and number of replicates used are indicated in the Figure legends. The unpaired two-tailed *t*-test was used to compare two groups, while ANOVA followed by Dunnett's or Tukey test was used to compare three or more groups. *P*-values < 0.05 were considered statistically significant.

**Reporting summary**. Further information on research design is available in the Nature Portfolio Reporting Summary linked to this article.

## Data availability

Uncropped immunoblots and TLC images are shown in Supplementary Figs. 4–11. The original data of the graphs and charts are shown in Supplementary Data 1. All other data provided in the article and Supplementary files are available from the corresponding author upon reasonable request.

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

## Acknowledgements
The authors would like to thank Rie Sugimoto for technical assistance and also acknowledge technical support from the Units of Molecular and Cellular Biology Research, Medical Bioresource Research, and Radioisotope Research, Central Research Institute, Kawasaki Medical School. The authors thank Editage (www.editage.com) for English language editing. This work was supported by the Japan Society for the Promotion of Science KAKENHI (grant numbers JP20K19732 to K.K. and JP20K11571 to K.T.); the Ichiro Kanehara Foundation (grant number 2020-23 to K.K.); Teraoka Scholarship Foundation (to K.K.); the Okayama Medical Foundation (to K.K.); Teijin Pharma Limited (grant numbers TJNS20200413014, TJNS20210412018, and TJNS20220411014 to K.K. and Y.O.); Kyowa Kirin Co., Ltd. (grant numbers KKCS20210402007 and KKCS20220401021 to K.K. and Y.O.); Bayer Academic Support (grant numbers BASJ20210402010 and BASJ20220401001 to K.K); Teijin Nakashima Medical Co., Ltd. (to K.K. and Y.O.); Otsuka Pharmaceutical Co., Ltd. (grant number AS2022A000441768 to K.K. and Y.O.); Asahi Kasei Pharma Corporation (grant number APJS20220406011 to K.K. and Y.O.); EA Pharma Co., Ltd. (grant number AS2022A000003641 to K.K. and Y.O.); Senko Medical Instrument Mfg. Co., Ltd. (to K.K. and Y.O.); CSL Behring K.K. (grant number AS2022A000009767 to K.K. and Y.O.) and Research Project Grants from Kawasaki Medical School (grant numbers R01S-002, R02S-002, and R04Y-001 to K.K.). The funders had no role in study design, data collection, decision to publish, or preparation of the manuscript.

## Author contributions
Conceptualization, K.K., N.U., and K.T.; Formal analysis, K.K.; Investigation, K.K., H.A., and R.K.; Data curation, K.K., and H.A.; Writing—original draft, K.K., Y.O., and K.T.; Writing—review & editing, K.K., H.A., R.K., Y.T., H.I., A.Y., N.U., T.T., Y.O., and K.T.; Visualization, K.K.; Project administration; K.K. and K.T.; Funding acquisition, K.K., Y.O., and K.T. All authors approved the manuscript for publication.

## Competing interests
The authors declare the following competing interests: K.K. and Y.O. received grants from Teijin Pharma Limited, Bayer Yakuhin, Ltd., Kyowa Kirin Co., Ltd., Teijin Nakashima Medical Co., Ltd., Otsuka Pharmaceutical Co., Ltd., Asahi Kasei Pharma Corporation, EA Pharma Co., Ltd., Senko Medical Instrument Mfg. Co., Ltd., and CSL Behring K.K. H.I. is an employee of Maruho Co., Ltd. All other authors declare no conflicts of interest.
