## [Transparent Peer Review File · Communications Biology]

Reviewers' comments:

Reviewer #1 (Remarks to the Author):

SIGNIFICANCE

This paper provides insights into the function of a Ca²⁺-dependent and endoplasmic reticulum associated lysophospholipase (GDE7). The findings of the authors are of major interest because this study shows that GDE7 is essential for the production of cyclic phosphatidic acid (cPA). This finding provides an exciting new connection with the existing knowledge that cPA suppresses the activity of nuclear peroxisome proliferator-activated receptor γ (PPAR γ) which is essential for mediator of adipogenic lineage differentiation and/or the metabolic functions of PPAR γ in phenotype-committed adipocytes and other cell types.

CRITIQUE

The biochemical findings in this paper are well designed and executed, with effective intercalation of findings obtained by in vitro or cell culture-based experimentation. The design of the study is rigorous, the data are of high quality and the scientific presentation is esthetically pleasing. The following points could enhance this strong study.

SPECIFIC POINTS

- 1) The authors show that "GDE7 deficiency derepresses the PPAR γ pathway in human mammary MCF-7 cells, suggesting that cPA functions as an intracellular lipid mediator. These findings lead to a better understanding of the biological role of GDE7 and its product, cPA" (Cited from Abstract). The selection of MCF-7 breast cancer cells as the vehicle for cell culture studies to establish physiological relevance of the GDE7/cPA mechanism in "living cells" is not the most exciting or optimal choice. Solid biochemistry becomes more interesting in a superb biological context. A far more interesting experiment that would dramatically increase the impact of this paper is if the authors could show that loss of GDE7 de-represses PPAR γ in mesenchymal stem cells to accelerate adipogenesis in the absence or presence of adipogenic stimuli. This experiment should be realistic with NIH 10T1/2 cells or 3T3-L1 adipocytes (from ATCC), or perhaps even more relevant, using either primary bone marrow MSCs from mice or human MSCs (e.g., from Lonza).
- 2) The proper gene symbol for PPAR γ is PPARG and it should minimally be used at the point of first mention. AI based text mining algorithms are more accurate with proper gene symbols thus increasing the potential for citation of the paper.

Reviewer #2 (Remarks to the Author):

This study asked whether GDE7 generates cPA in living cells and found that it does (COS and MCF7), although it is not surprising since it is known to generate other PA's. They also did structural/topological analyses to identify the active site (on the luminal side in the ER) and did mutagenesis biochemical work to find specific amino acid residues important for GDE7 catalytic activity.

Comments:

1. There is a great deal of basic biochemistry in this study with little biological rationale for conducting the investigation .
2. It is unclear why the reader would be interested in cPA , as it is relatively low in abundance in human tissue. The authors need to make the case for studying cPA and GDE7.
3. The biological meaning of the study was not addressed.

4. Concepts such as lipogenesis are at times confounded with adipogenesis.
5. The rationale for use of COS and MCF 7 cells is unclear , with little biological relevance to adipogenesis.
6. Is cPA necessary ?meaningfully different than LPA? Needs to be discussed and clarified.
7. Glucose homeostasis, cardiovascular disease, and cancer are mentioned but little is provided to connect these clinical concepts to the CPA and GDE7.
8. A prior study (Tsukahara et al, Molecular Cell 39, 421–432, August 13, 2010 , [https://www.cell.com/molecular-cell/pdf/S1097-2765\(10\)00569-1.pdf](https://www.cell.com/molecular-cell/pdf/S1097-2765(10)00569-1.pdf)) showed "Phospholipase D2-Dependent Inhibition of the Nuclear Hormone Receptor PPAR γ by Cyclic Phosphatidic Acid. " GDE7 is a lysophospholipase D enzyme family member (aka GDPD3) and therefore it is not surprising that there would be overlap in terms of an effect on PPAR γ . This manuscript fails to put their findings into context of what has already been shown.
9. This manuscript fails to show a convincing effect on PPAR γ in Fig 6 .
10. Listed PPAR γ targets are not conventional ones used in the literature .
11. Protein expression in addition to mRNA is needed in Fig 6 for PPAR γ targets.
12. Typical PPAR γ targets such as aP2 (Fabp4) and adiponectin are required.
13. In the Tsukahara et al study , they showed the PPAR γ agonist rosiglitazone upregulation of CD36 and CYP27a1 in 3T3L1 cells which bolstered these as targets of PPAR γ however they are not conventional targets used in the literature. Rosiglitazone does not upregulate Csf1. Csf1 is not a known PPAR γ target as presented in Figure 6.
14. Conclusions about adipogenesis cannot be made without investigation of adipogenic differentiation.

Reviewer #3 (Remarks to the Author):

The study shows for the first time the activity of hGDE7 in both living cells and cell-free systems. In addition, the authors also measured the differences in cyclic phosphatidic acid (cPA) products between the cell-free and living cell conditions. However, three following issues need to be solved to have the scientific attainments of GDE7.

1. Because all the experiments were repeated at least twice, would you please clarify the sample size (n = 3, n = 4) is from one time or if this is the combination of all the experimental replicates?
2. Since the cPA is also produced by Phospholipase D2, it is essential to have the experiment discriminate whether cPA products are from GDE7 activity or other pathways.
3. The previous study showed that the generation of N-acyl ethanolamines and lysophosphatidic acids is calcium-dependent. However, there is no evidence of a similar activity for the cPA generation. Therefore, another experiment is required to elucidate the novel mechanism of GDE7.

Author Response to Reviewers

We carefully revised our manuscript according to the suggestions of reviewers. We made a note in **yellow ink** where we added in the revised manuscript. Our point-by-point responses to the reviewer comments are as follows.

(Reviewer #1-1) The authors show that “GDE7 deficiency derepresses the PPAR γ pathway in human mammary MCF-7 cells, suggesting that cPA functions as an intracellular lipid mediator. These findings lead to a better understanding of the biological role of GDE7 and its product, cPA” (Cited from Abstract). The selection of MCF-7 breast cancer cells as the vehicle for cell culture studies to establish physiological relevance of the GDE7/cPA mechanism in “living cells” is not the most exciting or optimal choice. Solid biochemistry becomes more interesting in a superb biological context. A far more interesting experiment that would dramatically increase the impact of this paper is if the authors could show that loss of GDE7 de-represses PPAR γ in mesenchymal stem cells to accelerate adipogenesis in the absence or presence of adipogenic stimuli. This experiment should be realistic with NIH 10T1/2 cells or 3T3-L1 adipocytes (from ATCC), or perhaps even more relevant, using either primary bone marrow MSCs from mice or human MSCs (e.g., from Lonza).

(Response) Thank you for your helpful comments. As the reviewers pointed out, we agree that it is quite important to examine whether GDE7 activity affects PPAR γ -mediated adipogenesis in MSCs and adipocytes. According to the comments, we performed experiments using 3T3-L1 cells. Similar to the previous report studying PLD2 (Tsukahara et al, Molecular Cell 2010), we examined whether overexpression of GDE7 suppresses the PPAR γ pathway activated by the agonist rosiglitazone (ROSI) in these cells. The results showed that mRNA expression of PPAR γ downstream factors (Cyp27a1, Adipoq, and Fabp4) was suppressed by hGDE7 overexpression. These results suggest that cPA produced by GDE7 functions as a lipid mediator in various cell types. Recently, Shimizu et al showed that mRNA expression level of GDE7 was increased under hypoxic conditions in Caco-2 cells (Shimizu et al, Lipids 2023). Since the endogenous mRNA level of GDE7 was found to be low in 3T3-L1 cells in the present study, it is required to analyze GDE7 activators and transcription factors that regulate its expression in the future. Further studies using primary bone marrow MSCs are also needed to determine whether the suppression of PPAR γ by cPA produced by GDE7 is involved in the signaling that determines MSC differentiation into adipocytes (Li et al, Curr Stem Cell Res Ther 2018). We added these to Results (Fig 6) and Discussion sections.

Furthermore, it has been reported that activation of PPAR γ could inhibit growth and progression of malignant breast cancer (Augimeri et al, Cancers 2020). Thus, higher expression of GDE7 could suppress PPAR γ via increased cPA levels and potentially cause malignancy of breast cancer. We

also added this biological relevance in MCF-7 to Discussion section in the revised manuscript.

(Reviewer #1-2) The proper gene symbol for PPAR γ is PPARG and it should minimally be used at the point of first mention. AI based text mining algorithms are more accurate with proper gene symbols thus increasing the potential for citation of the paper.

(Response) Thanks for pointing this out. We corrected “PPAR γ ” as “PPAR γ (encoded by *PPARG*)” and “GDE4 and GDE7” as “GDE4 and GDE7 (also known as glycerophosphodiester phosphodiesterase domain-containing proteins GDPD1 and GDPD3, respectively).”

(Reviewer #2-1) There is a great deal of basic biochemistry in this study with little biological rationale for conducting the investigation.

(Response) We agree with the comment that the biological rationale for conducting the study should be described. We have modified the introduction as follows: LPA and cPA also modulate adipogenic differentiation and glucose homeostasis through nuclear peroxisome proliferator-activated receptor γ (PPAR γ) as an agonist and an antagonist, respectively. Although the biosynthetic mechanisms of LPA (Geraldo et al, Signal Transduction and Targeted Therapy & McIntyre et al, PNAS 2003) and another endogenous agonist, 15deoxy- $\Delta^{12,14}$ -prostaglandin J₂ (Forman et al, Cell 1995 & Kliewer et al, Cell 1995), are well established, those of cPA are not fully elucidated *in vivo*.”

To indicate the importance of the GDE7-cPA-PPAR γ pathway in breast cancer and adipocytes, we have also added figures to Fig. 6 and revised Results and Discussion sections according to the following points.

(1) It has been reported that activation of PPAR γ could negatively regulate growth and progression of malignant breast cancer (Augimeri et al, Cancers 2020), although it is not fully understood how cancer suppresses PPAR γ . Our results suggest that higher expression of GDE7 in MCF-7 cells could suppress PPAR γ via increased cPA levels and cause malignancy of breast cancer.

(2) PPAR γ is necessary and sufficient for adipocyte differentiation and is implicated in the transcription of a group of adipogenesis-specific genes, although the mechanism of its suppression is not fully understood. Since it is quite important to examine whether GDE7 activity affects PPAR γ -mediated adipogenesis in MSCs and adipocytes, we added experiments using 3T3-L1 cells to the present study. Similar to the previous report studying PLD2 (Tsukahara et al, Molecular Cell 2010), we examined whether overexpression of GDE7 suppresses the PPAR γ pathway activated by the agonist rosiglitazone (ROSI). We found that mRNA expression of PPAR γ downstream factors (Cyp27a1, Adipoq, and Fabp4) was suppressed by overexpression of hGDE7. These results suggest the physiological relevance of cPA produced by GDE7. Recently, Shimizu et al showed that mRNA

expression level of GDE7 was increased under hypoxic conditions in Caco-2 cells (Shimizu et al, Lipids 2023). Since the endogenous mRNA level of GDE7 was found to be low in 3T3-L1 cells in the present study, it is required to further analyze GDE7 activators and transcription factors that regulate its expression. Further studies using primary bone marrow MSCs are also needed to determine whether the suppression of PPAR γ by cPA produced by GDE7 is involved in the signaling that determines MSC differentiation into adipocytes (Li et al, Curr Stem Cell Res Ther 2018).

(Reviewer #2-2) It is unclear why the reader would be interested in cPA, as it is relatively low in abundance in human tissue. The authors need to make the case for studying cPA and GDE7.

(Response) Thank you for your comment. Because of the low concentration of lysophospholipids including cPA in human blood, their quantification using human samples is difficult and has not been studied extensively. Recently, methods for the quantification of lysophospholipids including cPA have been established (Li et al, Anal Bioanal Chem. 2023), and it is expected that the role of cPA in health and diseases will be clarified in the near future. Furthermore, lipid mediators are generally produced locally and act on their targets in the vicinity. Thus, even if the cPA amounts in the whole tissue are low, it is possible that the high local concentrations could exert physiological significance. Since it has been reported that cPA produced by ATX is converted to β -LPA (Sci Rep 2021), it is also possible that cPA is actually produced in much higher amounts despite that it has not been detected. We added the above information to Discussion section in the revised manuscript.

(Reviewer #2-3) The biological meaning of the study was not addressed.

(Response) Thank you for your comment. We showed that cPA produced by GDE7 suppresses the activity of PPAR γ , which is essential for differentiation and metabolic functions of adipocytes and other cells. The results provide the important biological significance to the research area of lipid mediators and PPAR γ . We added them to Discussion section in the revised manuscript.

(Reviewer #2-4) Concepts such as lipogenesis are at times confounded with adipogenesis.

(Response) Thank you for your comment. To avoid confusing lipogenesis with adipogenesis, we have rephrased “adipogenesis” to “adipogenic differentiation.”

(Reviewer #2-5) The rationale for use of COS and MCF 7 cells is unclear, with little biological relevance to adipogenesis.

(Response) Thank you for your comment. We have added experiments with 3T3-L1 cells to Fig 6 for adipogenesis, as described above in response to #2-1.

(Reviewer #2-6) Is cPA necessary ?meaningfully different than LPA? Needs to be discussed and clarified.

(Response) Thank you for your comments. It is unclear whether cPA acts in a similar or opposite manner to LPA. However, at least for PPAR γ , LPA works as an agonist (McIntyre et al, PNAS 2003) and cPA as an antagonist (Tsukahara et al, Molecular Cell 2010). We added this to Discussion section in the revised manuscript.

(Reviewer #2-7) Glucose homeostasis, cardiovascular disease, and cancer are mentioned but little is provided to connect these clinical concepts to the CPA and GDE7.

(Response) Thank you for your comments. The phrase “the pathogenesis of cancer and cardiovascular diseases via the PPAR γ pathway” has been revised to “the pathogenesis of cancer and obesity via the PPAR γ pathway,” reflecting the results of the experiments using 3T3-L1 cells.

(Reviewer #2-8) A prior study (Tsukahara et al, Molecular Cell 39, 421–432, August 13, 2010 , [https://www.cell.com/molecular-cell/pdf/S1097-2765\(10\)00569-1.pdf](https://www.cell.com/molecular-cell/pdf/S1097-2765(10)00569-1.pdf)) showed “Phospholipase D2-Dependent Inhibition of the Nuclear Hormone Receptor PPAR γ by Cyclic Phosphatidic Acid. “ GDE7 is a lysophospholipase D enzyme family member (aka GDPD3) and therefore it is not surprising that there would be overlap in terms of an effect on PPAR γ . This manuscript fails to put their findings into context of what has already been shown.

(Response) Thank you for your comment. It has been reported that cPA produced by PLD2 functions as an antagonist of PPAR γ (Tsukahara et al.). Therefore, we hypothesized that if cPA produced by GDE7 functions as a lipid mediator, its PPAR γ -suppressive action would also overlap with that of PLD2. From our results using GDE7-KO MCF7 cells and GDE7-overexpressing 3T3-L1 cells, we concluded that cPA produced by GDE7 also functions as a suppressor of PPAR γ signaling.

(Reviewer #2-9)

This manuscript fails to show a convincing effect on PPAR γ in Fig 6 .

(Response) Thank you for your comment. We performed additional experiments using 3T3-L1 cells and found the suppression of PPAR γ by cPA produced by GDE7 in these cells. Together with the results of MCF-7 cells, we think that we have now shown the convincing effect on PPAR γ .

(Reviewer #2-10)

Listed PPARg targets are not conventional ones used in the literature .

(Response) Thank you for your comment. Conventional PPAR γ targets such as Adipoq (adiponectin) and Fabp4 (aP2) were also evaluated in experiments with 3T3-L1 cells, and these results were added to Fig 6 in the revised manuscript.

(Reviewer #2-11)

Protein expression in addition to mRNA is needed in Fig 6 for PPARg targets.

(Response) Thank you for your comment. We examined protein expression of PPAR γ downstream factors in 3T3-L1 cells. When 3T3-L1 cells were treated with 1 μ M ROSI for 24 h under serum-free conditions, we could detect mRNA but not protein expression of the PPAR γ downstream factors (Figure for reviewers, **a**). When 3T3-L1 cells were treated with 1 μ M ROSI in the presence of 10% calf serum for 1 week (medium change every other day), protein expressions of the PPAR γ downstream factors were detected. However, overexpression of hGDE7 hardly suppressed the expression of these proteins (Figure for reviewers, **b**). This could be due to the influence of other lipid mediators in the calf serum under these conditions. Although we were unable to detect an effect of cPA produced by GDE7 on the protein expression of PPAR γ downstream factors in our additional experiments, the conclusion of the present study is not changed. However, the role of GDE7-cPA in adipogenic differentiation needs further investigation.

(Reviewer #2-12)

Typical PPARg targets such as aP2 (Fabp4) and adiponectin are required.

(Response) Thank you for your comment. mRNA expression of typical PPAR γ targets such as Adipoq (adiponectin) and Fabp4 (aP2) were also evaluated in additional experiments with 3T3-L1 cells, and these results were added to Fig 6 in the revised manuscript.

In experiments with MCF-7 cells, mRNA expression of these genes was not detectable.

(Reviewer #2-13)

In the Tsukahara et al study , they showed the PPARg agonist rosiglitazone upregulation of CD36 and

CYP27a1 in 3T3L1 cells which bolstered these as targets of PPAR γ however they are not conventional targets used in the literature. Rosiglitazone does not upregulate Csf1. Csf1 is not a known PPAR γ target as presented in Figure 6.

(Response) Thank you for your comment. We have withdrawn the Csf1 data because the increased expression of Csf1 may be due to cPA-independent GDE7 action.

(Reviewer #2-14) Conclusions about adipogenesis cannot be made without investigation of adipogenic differentiation.

(Response) Thank you for your comment. We added experiments with 3T3-L1 cells and found that GDE7-cPA also suppresses PPAR γ downstream factors in ROSI-induced adipogenic differentiation.

(Reviewer #3-1) Because all the experiments were repeated at least twice, would you please clarify the sample size (n = 3, n = 4) is from one time or if this is the combination of all the experimental replicates?

(Response) Thank you for your comment. We noted that the sample size in each experiment is from one time in the revised manuscript.

(Reviewer #3-2) Since the cPA is also produced by Phospholipase D2, it is essential to have the experiment discriminate whether cPA products are from GDE7 activity or other pathways.

(Response) Thank you for providing these insights. To investigate whether cPA products are from GDE7 activity or other pathways, we tested cPA-producing activity of hGDE7 in the presence of the PLD1/2 inhibitors FIPI and BML279 and the ATX inhibitor S32826. We found that these inhibitors did not reduce cPA production by hGDE7. In contrast, BrP-LPA, which inhibits the FS-3 degrading activity of GDE7 (Kitakaze et al, J Lipid Res 2021), reduced cPA production by approximately 20%. These results suggest that GDE7 and PLD2/ATX act separately. We have added these results to the supplementary Figure 1 in the revised manuscript.

(Reviewer #3-3) The previous study showed that the generation of N-acyl ethanolamines and lysophosphatidic acids is calcium-dependent. However, there is no evidence of a similar activity for the cPA generation. Therefore, another experiment is required to elucidate the novel mechanism of GDE7.

(Response) Thank you for your suggestion. We investigated cPA production by GDE7 with or without Ca^{2+} and in the presence of the calcium chelator EGTA. Similar to previous results for lysophosphatidic acids and N-acylethanolamines, the generation of cPA by GDE7 was Ca^{2+} -dependent. These results were added to the supplementary Figure 1 in the revised manuscript.

REVIEWERS' COMMENTS:

Reviewer #1 (Remarks to the Author):

The authors have adequately addressed the comments from all reviewers and added additional data to strengthen the work.

Reviewer #2 (Remarks to the Author):

Summary: The authors have mostly addressed the initial review comments.

1. Addressed

2. Recommend to revise.

- The authors make the case this investigation is irrelevant for cancer and obesity (based on in vitro data) but fail to convincingly show effect on adipogenic differentiation of relevant stem cells, nor do they show an effect on tumorigenesis.
- Obesity needs to be removed as a clinical correlate. For breast cancer, it is ok to speak of an in-vitro model of breast cancer in which a pathway implicated in tumor progression is attenuated, but some care is needed with regard to use of the general overarching term 'cancer'.

3. Addressed.

4. Addressed.

- It is helpful to have the 3T3L1 pre-adipocytes.

- Still need to explain why COS were used.

5. Addressed

6. Not addressed. Glucose homeostasis, cardiovascular disease, and cancer are broad terms that do not belong in the manuscript. It is possible to discuss MCF7 as a human breast cancer cell line (as above).

7. Addressed

8. Addressed- Helpful addition of 3T3L1 experiment and discussion section noting the need for MSC for more rigorous investigation of adipogenesis.

9. Addressed

10. Incompletely addressed. They did a nice job of performing this additional experiment but need to include it fully in the main figures, instead of only including the positive data. This should not be in a figure for reviewers or in a supplemental figure. Negative data is critical to improve scientific transparency. If the term adipogenesis remains in the manuscript, this additional figure needs to be incorporated.

They can discuss the negative (protein) data. For example: Pre-adipocytes are not as multipotential so it might be harder to show a suppression of adipogenesis in these cells. Also rosiglitazone is a fairly powerful driver of adipogenesis. It is possible that with other types of adipogenic media, a different result could be obtained. They can discuss this.

11. Addressed

12. Addressed

Reviewer #3 (Remarks to the Author):

The authors provided satisfactory responses to my queries and conducted additional experiments. As a result, I am in agreement with their responses and find their revised manuscript to be suitable.

Author Response to Reviewer

We carefully revised our manuscript according to the suggestions of reviewers. We made a note in **blue ink** where we added in the latest revised manuscript. Our point-by-point responses to the reviewer comments are as follows.

(Reviewer #2-2)

Recommend to revise.

- The authors make the case this investigation is relevant for cancer and obesity (based on in vitro data) but fail to convincingly show effect on adipogenic differentiation of relevant stem cells , nor do they show an effect on tumorigenesis.
- Obesity needs to be removed as a clinical correlate. For breast cancer, it is ok to speak of an in-vitro model of breast cancer in which a pathway implicated in tumor progression is attenuated, but some care is needed with regard to use of the general overarching term ‘cancer’.

(Response) Thanks for pointing this out. We carefully withdrew the expressions “cancer” and “obesity” in the discussion of our results.

(Reviewer #2-4)

Addressed.

- It is helpful to have the 3T3L1 pre-adipocytes.
- Still need to explain why COS were used.

(Response) Thank you for your comments. COS-7 cells are commonly used for protein expression, and we added this to Results section in the latest manuscript.

(Reviewer #2-6)

Not addressed. Glucose homeostasis, cardiovascular disease, and cancer are broad terms that do not belong in the manuscript. It is possible to discuss MCF7 as a human breast cancer cell line (as above).

(Response) Thank you for your comment. We corrected the manuscript as described above in response to #2-2.

(Reviewer #2-10)

Incompletely addressed. They did a nice job of performing this additional experiment but need to include it fully in the main figures , instead of only including the positive data. This should not be in a figure for reviewers or in a supplemental figure. Negative data is critical to improve scientific

transparency. If the term adipogenesis remains in the manuscript, this additional figure needs to be incorporated.

They can discuss the negative (protein) data. For example: Pre-adipocytes are not as multipotential so it might be harder to show a suppression of adipogenesis in these cells. Also rosiglitazone is a fairly powerful driver of adipogenesis . It is possible that with other types of adipogenic media, a different result could be obtained. They can discuss this.

(Response) Thank you for your comment. We included the protein data as a new panel in Fig. 6 (Fig. 6c) and added its description to Results section. Corresponding antibody information and uncropped gel images were also added to the Methods section and Supplementary information, respectively.